# Air Pollution Health Literacy among Active Commuters in Hamilton, Ontario

**DOI:** 10.3390/ijerph20136282

**Published:** 2023-07-03

**Authors:** Reed Ciarloni, K. Bruce Newbold

**Affiliations:** School of Earth, Environment & Society, McMaster University, Hamilton, ON L8S 4K1, Canada

**Keywords:** active transportation, active travel, air pollution, public health, AQHI, environmental health literacy, air pollution health literacy, health belief model

## Abstract

The promotion of active transportation (AT), which has been broadly defined as a physical effort performed by the traveller to produce motion, has been a popular strategy to reduce vehicular emissions, improve air quality, and promote physical activity. However, individuals who engage in AT may incur increased exposure to air pollutants and thus potential health impacts. This research sought to determine how active commuters understand the health risks associated with air pollution during their commutes, and whether they engage in any behaviours to limit exposure. An online survey was adapted from the Environmental Health Literacy framework to assess air pollution health literacy among active commuters in Hamilton, ON, and generated a sample size of 192 AT users. Analyses involved the use of frequency tables and cross-tabulations for the quantitative data, and the Health Belief Model and thematic analysis to interpret the qualitative data. Results revealed that most AT users do not adopt behaviours that would limit air pollution exposure on commutes and exhibited low self-rated knowledge of the health impacts of air pollution exposure. Issues of perceived susceptibility and severity, barriers, cues to action, and self-efficacy all further impacted the likelihood of adopting health protective behaviours. Conclusively, air pollution is an often-neglected consideration among active commuters, with air pollution knowledge predicting the likelihood of behavioural modification.

## 1. Introduction

Anthropogenic effects on the environment, such as industrialization and urbanization, have contributed to increases in pollution and a subsequent poor air quality. Pollution-related deaths have increased by 66% in the last 20 years, accounting for 6.7 million deaths per year worldwide [1]. The transportation sector produces approximately 70% of environmental pollution due to the variety of pollutants that make up traffic emissions, both primarily [2,3] and secondarily through chemical reactions with other pollutants in the atmosphere.

The promotion of active transportation (AT) has been a popular strategy among developed nations looking to reduce vehicular emissions. The definition of AT has evolved since the inception of the term to reflect the growing number of modalities, and the need for inclusivity around what can be considered as AT. This is reflected in the formal definition of AT, which states that active travel is considered “Travel in which the sustained physical exertion of the traveller directly contributes to their motion” [4] p. 154. The application of AT has also found its use in public health, offering a co-benefit for environmental and public health concerns. AT initiatives in public health seek to target the high rates of non-communicable diseases (NCDs) and low participation in physical activity that is prevalent in Western societies. NCDs of concern typically include cardiovascular disease, coronary heart disease, stroke, and certain cancers [5,6,7,8].

However, the ability to reap the environmental and health benefits of AT is reliant on a variety of factors. Specifically, the likelihood of adopting AT behaviours is contingent on features of the urban environment and transportation planning, and a variety of social, economic, social, and environmental factors [9,10]. Due to this, active commuters may experience several safety and health-related issues during AT. This research is particularly concerned with air pollution exposure during AT. Individuals who engage in strenuous outdoor exercise are at risk for incurring greater health risks when exposed to ambient air pollution [11]. Current research supports that individuals who use AT may be at an increased risk of exposure to air pollutants through high inhalation rates [12], proximity [13], duration [14], and the time [15] of their commute that can result in confounding health impacts. However, few studies have explored the extent to which active commuters understand the health risks that are associated with exposure during their commutes, and if active commuters engage in efforts to mitigate air pollution exposure. This research seeks to contribute to the literature regarding commuter knowledge of air pollution exposure by assessing environmental health literacy (EHL) among active commuters, with the primary objective of determining whether active commuters understand the air pollution health risks incurred during AT, and if they engage in health-protective behaviours to mitigate their exposure. Research on this topic is necessary during the transitionary period from vehicular to more sustainable modes of transportation to provide evidence-based decision making for governments, policymakers, and community organizations on how air pollution is understood by this at-risk group to maximize the health outcomes of active commuters in the interim.

Section 2 addresses the current state of knowledge surrounding AT, air pollution exposure, and how air pollution affects the commuter’s health. This section also introduces the concept of EHL and its applicability for assessing air pollution health literacy. Section 3 describes the locale of research focus and the methods and materials utilized to assess air pollution health literacy. Data was collected via online questionnaires, and asked AT users’ questions pertaining to transportation, air pollution and health knowledge, and demographics/socioeconomic status. The results, presented in Section 4, were interpreted using descriptive statistics and cross-tabulations. Qualitative responses required thematic analysis using the Health Belief Model.

## 2. Background

To effectively communicate the relationship between air pollution exposure during AT and its potential health impacts on active commuters, an extensive review of the current literature has been provided. The inclusion of this review was warranted as it articulates true differences in air pollution exposure and the subsequent health impacts as a result of AT. The aim of this section was to provide the reader with a substantive overview as to why air pollution exposure on commutes is significant, and how the emerging EHL framework can serve as an important reference for assessing air pollution health literacy.

### 2.1. Air Pollution Exposure on Commutes

AT offers a promising solution for mitigating vehicular pollution. Research by Woodcock et al. [16] postulated that large-scale adoption of AT modalities could result in meaningful reductions in transit-related emissions of one-half by 2030. Even moderate use of AT has shown to be remarkably effective in improving air quality, given that urban trips by motor vehicles yield high-per-kilometer emissions. Substituting short-distance urban travel with AT could generate substantial benefits with respect to air quality [10]. While the benefits of AT are not to be undermined, the application and efficacy of AT policy and its adoption among the public are highly contingent on factors of the built environment. Reynolds et al. [17] identified factors of the built environment that are integral to AT, such as connectivity, accessibility, safety, and bike/pedestrian-friendly infrastructure. Similarly, Glazener and Khreis [18] detailed the importance of specific factors, such as sprawl, which characterizes current urban growth and development, as further reinforcing car dependency and impairing critical connectivity. Of note, increases in AT do not necessarily translate to complete substitution of motorized vehicles, thereby making it difficult to quantify the impact of AT on air pollution reduction [17].

Without a significant adoption of AT among the public, the benefits of AT become less impactful, and individuals who engage in AT may be at increased risk of exposure and inhalation of air pollutants. Engström and Forsberg [19] suggested that urban areas are the most conducive to AT but introduce substantial exposure to air pollution. They found active commuters can be exposed to higher concentrations of certain air pollutants by up to five times compared to motorists during rush hour. Here, the variables of time and place are important when quantifying exposure. A study conducted by Dirks et al. [15] on air pollution exposure in school walking routes found evidence to support a time-sensitive association between pollution exposure and commuting. Individuals who commuted on the busier side of the road within the morning hours incurred a higher risk of exposure than those who walked on less congested streets. Moreover, as commutes to work or school often occur during rush-hour, this time of commuting can lead to an increased pollution exposure from the proximity to, and amount of, vehicular traffic [13]. Additionally, the duration of active commutes carries a greater burden of exposure to air pollution than for individuals who use motor vehicles. Ramos et al. [12] articulated the importance of commute time through increased inhalation doses, noting that longer commute durations were highly correlated with higher inhalation doses of air pollutants. Sommar et al., [20] confirmed that there are true variations in inhalation doses between motorists and active commuters supported by the proximity to traffic and a high exchange rate of air during active commutes. Quantifiable differences in these inhalation rates found that cyclists’ inhalation rates of NO_2_ were three times than that of motorists [21].

### 2.2. Health Impacts Associated with Air Pollution

Air pollution has been associated with an increase in both morbidity and mortality due to its known association with NCDs. Commonly, air pollution-related health conditions present short-term and long-term consequences. The short-term effects of ambient air quality can aggravate pre-existing conditions and cause acute symptoms ranging in severity from mild irritation of the respiratory tract and difficulty in breathing to more serious conditions, such as asthma, pneumonia, bronchitis, and lung and heart problems [22]. Sustained exposure to air pollutants can result in long-term health impacts, such as respiratory diseases, cardiovascular diseases, and certain types of cancers [22]. The varying health impacts introduced by air pollution are formed as the result of a complex make-up of particles and gases in the atmosphere which have the propensity to interfere with biological processes. Indeed, exposure to pollutants, including particulate matter (PM, inclusive of both PM_10_ and PM_2.5_), ozone (O_3_), nitrogen oxide (NO), nitrogen dioxide (NO_2_), volatile organic compounds (VOCs), sulphur dioxide (SO_2_), and carbon monoxide (CO) have all been associated with a range of negative health outcomes, including irritation of the throat, nose, and eyes [23], cancer [22], asthma, COPD, cardiovascular disease [24,25,26], and premature mortality [27,28,29,30], among other impacts.

### 2.3. Health Impacts Due to Air Pollution Exposure during Active Travel

It has been argued that a risk-benefit trade-off exists between AT and health when accounting for air pollution. Certainly, physical activity through AT has revealed positive outcomes, even when controlling for air quality. However, methodological limitations of this evidence have been noted, as ambient air quality has great variability in outdoor environments and the complex make-up of ambient air pollutants, and their fluctuating daily concentrations make it difficult to establish true beneficial health outcomes [14]. In contrast, a wealth of research supports a relationship between an increased burden of air pollution exposure and their adverse health outcomes among active commuters. Exposure to ambient air pollution during AT may elevate the risk of morbidity and mortality, rather than improve or prevent the NCDs that the public health agendas seek to address. A significant variable of air pollution exposure in active commuters results from transportation infrastructure and policy. Rojas-Rueda [31] have explained that there is often little interplay between AT policies and health policies, thereby creating the potential for health risks among the active commuters.

A systemic review and meta-analysis on exercise and air pollution found evidence to support a relationship between activity and air quality, which resulted in a poor cardiopulmonary and immune function, and exercise performance [32]. Similarly, traffic-related air pollution and neighbourhood walkability in Ontario, Canada was investigated to address the health impacts introduced by traffic-related air pollution. It was found that as traffic-related pollution increased, the potential health gains of the neighbourhood walkability decreased, particularly when controlling for cardiovascular disease risk factors, such as diabetes and hypertension [33]. Moreover, the association between activity duration is an important measure when seeking to establish the effects of AT and health. Pasqua et al. [14] found evidence of a dose response, noting that exercising in polluted urban environments could produce adverse health effects with respect to inflammatory and cardiovascular responses. Their findings suggested that activity benefits have a ceiling. When activity exceeded one hour in duration, the positive health effects of exercising in polluted environments significantly diminished as a consequence. Unequivocally, this relationship was also found to be true for short-term exposure. Short-term exposures to traffic air pollution through physical activity can produce adverse physiological responses with respect to cardiovascular morbidity in healthy young women [34]. While exposure to air pollution does not exclusively happen during commutes, Engström and Forsberg [19] stated that almost one-quarter of annual inhaled doses among active commuters in Stockholm were associated with vehicle emissions, which translated to an average of a 2.5% increase in premature death.

### 2.4. Environmental Health Literacy

The health impacts of air pollution and its significance to AT have been well researched. However, there is limited research on how these health impacts are understood by active commuters, and whether active commuters are knowledgeable of and utilize available air quality health resources. Thus, evaluating environmental health literacy is paramount to understanding how effective and transferable this research is in informing the behaviours of active commuters. Environmental Health Literacy (EHL) is a newly emerging framework that seeks to conceptualize public comprehension and utilization of environmental health and risk information, and the skills and actionable steps required for engaging in health-protective behaviour [35]. Often, individual-level competencies have been used to understand how the public engages with and incorporates EHL into their daily practices to form health-protective behaviours [36]. In a systemic review of EHL, Gray et al. [35] established that individuals have a general awareness of the health impacts introduced by the environment. However, there was a great deal of misinformation and a lack of critical information for reducing the exposure. This was corroborated in Fin and O’Fallon’s [37] work, where researchers found that participants could identify, with a fair degree of accuracy, the knowledge required of EHL. However, they were not as successful in identifying the actionable steps required. These findings are significant, as they reveal gaps in knowledge that impact risk-perception. Lindsey et al. [38] further emphasised the significance of EHL in addressing the perceptions of risk and safety, calling for EHL communication that is action-oriented and comprehendible among the lay public to ensure its efficacy.

### 2.5. Air Pollution Health Literacy

The efficacy of EHL in evaluating individual-level comprehension of environmental health hazards makes it a favourable framework for gauging knowledge of ambient air quality and health protective behaviours among active commuters. Tainio et al. [39] noted that individual perceptions of air quality and whether this information informs activity outdoors is uncertain and remains an under-researched area of study. Attempts to address air pollution perceptions have used an adapted EHL framework to gauge Ambient Air Pollution Health Literacy (AAPHL) among Taiwanese residents. This study was the first of its kind to employ a standard measure for understanding AAPHL and evaluated competency based on four matrices: access, understand, appraise, and apply. Notably, evidence from this study suggested that individuals have the greatest difficulty with accessing and appraisal of information, while the application of information correlated the least with the other matrices [40]. The idea of appraisal and application are important metrics, as they determine the ability of an individual to identify their health risks and employ health-protective behaviours in response to the perceived risk. Ramirez et al. [41] further confirmed these findings, noting the existence of significant disparities in the quality of air pollution communication and public knowledge, can affect how individuals engage in actionable health-protective behaviours. Critical gaps in air pollution health communication also result from poorly executed health information. Brugge et al. [42] used the EHL framework to produce traffic pollution fact sheets for a focus group of Puerto Ricans living in Boston, Massachusetts, near major roadways. They identified that EHL-reporting materials that the researchers thought would be appropriate for public education were not comprehensible among the members of this focus group. Importantly, effective knowledge translation can remedy this gap. When evaluating the factors that influence AT decisions among German commuters, Koenigstorfer [43] discovered that effective and comprehendible communication of air pollution along roadways could encourage health-protective behaviours among active commuters. 

## 3. Materials and Methods

### 3.1. Case Study: Hamilton, ON

The City of Hamilton has been defined by its heavy industrial presence. This place-based association of air pollution is important to consider, as it can greatly inform an individual’s perceived susceptibility and severity of air pollution on their health. This characteristic, in addition to a significant air pollution-related mortality rate, which was reported as 67 per 100,000 in 2016 [30], justifies Hamilton as an interesting contextual basis for studying air pollution health literacy among active commuters. Sources that contribute to air pollution levels and the distribution of air pollution throughout Hamilton are the result of a range of industrial, vehicular, topographic, meteorological, and atmospheric factors [44]. Importantly, while Hamilton’s industrial sector does contribute to local emissions, its impact is not bound by its location. The northeast and southwest winds cause pollution to concentrate in areas of the city that are far from the industrial sector. Wallace et al., [45] further confirmed that while the industry is commonly considered a main contributor of emissions in Hamilton, vehicular emissions play a more significant role in the city’s air quality. Air quality research conducted within the City of Hamilton corroborate that those who live in close vicinity to Hamilton’s industrial sector had poorer perceptions of air quality, while those that live on the southwest mountain had significantly higher positive air quality perceptions [46,47,48].

Moreover, the methods Hamiltonians use to inform their air quality decisions are variable and not always reliable. A survey conducted in Hamilton’s North End by Elliot et al. [49] revealed that Hamiltonians engage with air quality health messages through various media sources, with some relying on sensory input to determine air pollution levels. When seeking to determine knowledge of the air quality index (AQI), the majority of respondents (81%) were found to be familiar with the term, while 50% recognized it as a measure of air pollution or air quality, and 11% did not know what AQI stood for. 

In 2015, the Air Quality Health Index (AQHI) replaced the AQI to better capture the impact of air quality on health [50]. The AQHI is a comprehensive tool for assessing daily health risks introduced by air pollution and suggests behaviour modifications for reducing exposure. As of 2016, only 32% of Hamiltonians reported using the AQHI, and many respondents were not able to identify the critical air pollutants included in the AQHI, such as NO_2_, O_3_, and PM_2.5_, respectively [51]. A study on the factors influencing health behaviours in response to the AQHI in Hamilton, Ontario, identified barriers to consulting air quality health messages, which involved time constraints and uncertainties on where to check the AQHI information, misinformation, and personal contextual information that could skew air quality perceptions [52]. Importantly, while 75% of Hamiltonians understood that ambient air quality adversely impacts health, only 22% reported modifying their behaviour in response to perceived health risks [46,51]. 

While the role out of AQHI data and its use was initially limited, it is now readily accessible through local forecasts on Canada’s Weather Network (https://www.theweathernetwork.com/ca (accessed on 26 June 2023)) as well as the Government of Canada (https://weather.gc.ca/airquality/pages/onaq-009e.html (accessed on 26 June 2023)). Other real-time air quality information that draws upon the Ministry of Environment, Conservation and Parks (MECP) monitoring, and the Hamilton Air Monitoring Network (HAMN) is available at http://newreporting.hamnair.ca/ (accessed on 26 June 2023)). In addition, these web resources offer insights into understanding the AQHI value, their health impacts, and remedial action that a user can take to protect their health.

### 3.2. Study Instrument

This research employed a 40-item questionnaire administered anonymously online through LimeSurvey (The survey is available from the first author on request). The focus of the survey was to understand air pollution health literacy among active commuters through questions concerning knowledge and behaviour in response to air pollution exposure on commutes. The survey questions were both quantitative and qualitative, when necessary, and were grouped into three broad categories, which included transportation characteristics, demographic and socioeconomic characteristics, and air quality, health knowledge, and behavioural questions. The survey questions were developed in accordance with Environmental Health Literacy’s 3 key dimensions which are as follows: (1) awareness, knowledge, and understanding, (2) skills and self-efficacy, and (3) change and action [35], to evaluate air pollution health literacy. The Health Belief Model was utilized for survey development, and to achieve further depth and analysis of the questions concerning behaviour motivations.

### 3.3. Sample Population

Eligibility required three criteria, which were as follows: (1) current residence in Hamilton, (2) over the age of 18, and (3) use of AT as the primary mode of transport. In the context of this study, “primary” has been defined as more than 50% of an individual’s daily trips/commutes being active, and where AT is used four or more days of the week. The defined 50% was inclusive of any travel distance, time, chain trips, or any combination of travel modes, so long as AT accounted for the most frequently used method of transportation. Modes of AT can include any human-powered form of transport, such as walking, biking, scooter (non-motorized), etc. Moreover, these criteria were not specific to commuting destination, thereby allowing for the participation of individuals who primarily use AT to commute from school, leisure, errands etc. Conditions were set within the survey tool to ensure that the target population was represented in the survey results. Individuals that selected “no” to any of the above eligibility criteria, would either receive an alternative survey, or the survey would terminate. This was necessary to ensure that participants who did not identify as a primary AT user, would not be represented in the questions that were used to assess air pollution health literacy among AT users. In total, the survey generated 235 responses through voluntary response sampling. Of these, 218 were fully completed, and 192 were AT users.

Participants were recruited using a variety of methods to ensure a diverse and robust sample. Recruitment posters were distributed in various parks, trails, pedestrian streets, and university/college campuses throughout Hamilton to recruit individuals from different areas of the city, of different age groups, and of different socioeconomic conditions. Multiple bike shop owners and aligned organizations/groups were contacted to assist in sharing these recruitment materials, either by physical posters within bike shops, or E-poster sharing on social media and in monthly newsletters. These specific business/organizations/groups were chosen for survey distribution due to their association with active transportation, sustainability/environment, and air pollution, thus increasing the odds of our survey reaching the target audience. Lastly, online/social media advertisements were created with a direct link to the survey and the target audience was selected for through keywords in the ad details. Prior to the formal survey period, the survey tool was evaluated several times from a participant’s perspective to ensure that it ran properly. The survey became available to the public on 15 September 2022, and ended on 14 November 2022. This research project was approved by the McMaster Research Ethics Board.

## 4. Results

The data from the survey was analyzed quantitatively, with a few qualitative analyses performed on type-in responses. Qualitative responses added to the depth and breadth of the behavioural responses, allowing participants to expand on their behavioural reasoning or beliefs about air quality. The quantitative data was presented using descriptive statistics and includes the following sections: transportation characteristics (Section 4.1), socioeconomic and demographic characteristics (Section 4.2) of the sample, and results of the survey questions relating to air pollution knowledge, concern, and behaviour (Section 4.3 and Section 4.4). A series of cross-tabulations was presented in Section 4.5 to determine whether there was an association between the self-rated air pollution and health knowledge and the likelihood of engaging in health protective behaviour. Qualitative responses were analyzed using thematic analysis (Section 4.6, Section 4.7, Section 4.8, Section 4.9 and Section 4.10), a common analytical tool used in qualitative research that involves coding and organizing data into themes [53]. The Health Belief Model was utilized to explain the factors that influence the likelihood of engaging in health-promoting behaviour [54]. This research utilized the Health Belief Model detailed by Champion and Skinner [55].

### 4.1. Transportation Characteristics

Table A1 displays the transportation characteristics of the survey sample, with 192 participants identifying themselves as regular AT users. The majority of the sample (72%) had a driver’s license; however, the availability of a vehicle showed more variation, with “all the time” (24%) and “never” (21%), representing the highest responses. The format for the reporting mode of AT allowed for the “check all that apply”, producing the following percentages: walking represented 78% of responses, followed by biking (49%) and other AT modes (6%). Motivations for using AT offered a similar “check all that apply” question type, with health being the most common motivation for engaging in AT (76%). This was followed by environment (66%), proximity (63%), economic (63%), preference (54%), lack of transportation options (28%), and other (9%), respectively. More than two-thirds of the sample (126 responses or 66%), stated using AT between 10–12 months of the year. Participants were asked about their use of AT during varying weather conditions. Specifically, 161 (84%) reported using AT when it is extremely hot, while 73% reported using AT when it rained, respectively. The commuting environment for the participants equally represented main street/busy roads at 147 responses (77%) and side streets with 148 responses (77%), respectively. Less common responses included trails (30%), back roadways (8%), and other (4%). Active commuting purposes were overrepresented in leisure activities at 165 responses (86%), errands with 163 (85%), and work at 110 (57%), respectively. As for the average total daily commute time, over 35 min represented the highest response at 31%, followed by 15–25 min at 29%, and 25–35 at 24%, respectively. Lastly, when asked if AT was an important part of their lifestyle, 85% of participants selected yes.

### 4.2. Demographic and Socioeconomic Characteristics

Table A2 displays the demographic and socioeconomic (SES) profile of the survey sample. Most survey respondents identified as female, accounting for 63% of responses. Males represented 26% of the sample, while non-binary individuals totaled 5%, respectively. As for age, 65% of the sample were between the ages of 18 and 34, followed by 35–44 (17%), 55–64 (7%), and 45–54 (7%), respectively. One respondent reported that they were over the age of sixty-five. The most common response concerning race was white, accounting for 76% of the sample. The most common levels of education of participants were “bachelor’s degree” (38%) and “master’s degree or higher” (24%). Mirroring the educational profile of respondents, 22% reported a yearly income of over $125,000. Finally, most participants reported working full-time (43%), followed by student (21%), and part-time (15%), respectively. 

### 4.3. Air Quality Knowledge and Concerns

The responses to questions that concerned air pollution knowledge, concern, and behaviour are displayed in Table A3. Participants were asked to rate their knowledge, awareness, and concern of the health risks of air pollution. Unequivocally, the majority of participants reported their levels of knowledge being between somewhat knowledgeable and not very knowledgeable across knowledge-based questions. Rankings that represented confidence in knowledge or a high degree of knowledge concerning air pollution’s effect on health were marginally reported. For instance, only 19% of active commuters reported being knowledgeable or very knowledgeable of the long-term effects associated with air pollution exposure (Q10). Sixty-five percent of respondents reported believing the positive aspects of AT outweigh the potential negative effects of air pollution, and twenty-eight percent stated uncertainties (Q5).

The occurrence of a low self-rated knowledge could be explained by the perceived availability of air quality in health information. Figure 1 displays the responses regarding where to find air quality and health information. Remarkably, 50% of survey respondents stated not knowing where to seek information about in air quality and health. Not knowing where to find air quality and health information can provide further context to the responses displayed in Figure 2, where most participants stated being only somewhat (50%) or not very concerned (27%) about the health risks associated with air pollution. Importantly, Figure 3 articulates a strong consensus among the active commuters, with 85% of participants either agreeing or strongly agreeing that information on the health impacts of air pollution exposure during commutes should be made more available to the public.

### 4.4. Air Quality and Behaviour

To determine whether active commuters engage in behaviours that would mitigate air pollution exposure, participants were first asked how frequently air quality informs their active transportation decisions. The results of these survey responses reveal that air quality is generally an insignificant factor in AT decisions. Indeed, this notion has been reflected in Figure 4, where active commuters reported nearly unanimously that air quality rarely (56%) or never (33%) informs their AT decisions. Marginal air quality considerations on commutes were found to correspond with an equally low reporting of AQHI checking. Figure 5 illustrates that 91% of participants stated that they do not check the AQHI daily prior to their commutes. While consulting air quality and health reporting tools, such as the AQHI is the recommended behaviour to mitigate air pollution exposure, the questionnaire sought to determine what behaviours, if any, are being adopted if not checking the AQHI. As such, participants were asked if they engage in any behaviours that limit exposure to air pollution. To this question, 75% reported not engaging in any behaviours that would limit their exposure to air pollution on their commute, as shown in Figure 6. Notably, there were variations between AQHI checking, where 91% reported “no” and if any behaviours are taken to limit exposure at 75%, respectively. This variance was expanded with qualitative responses and is reported in the latter section.

### 4.5. Cross-Tabulations

Results of the descriptive statistics revealed low self-rated knowledge and a low adoption of health protective behaviours in response to air pollution on commutes. Additional analysis in the form of cross-tabulations was performed on the data to determine how knowledge affects behaviour.

Table 1 shows that AQHI checking was found to have a statistically significant association with engaging in behaviours that limit exposure to air pollution on commutes (*p* = 0.000). Seventy-nine percent of participants that responded “no” to checking the AQHI also reported not engaging in any behaviours that would limit their air pollution exposure on commutes. Whether or not the participants knew where to find air quality health information was also found to be significantly associated with self-rated knowledge of the positives and negatives associated with active transportation and air pollution. Table 2 shows the results of this relationship, suggesting that participants that reported “no” to knowing where to seek information regarding in air quality and health were more likely to represent lower scores of self-rated knowledge, with the majority between one and three. Individuals that selected “yes” to knowing where to seek information regarding in air quality and health trended towards higher self-rated knowledge scores of 4 or 5 (*p* value = 0.027). The last significant association found in Table 3 concerns whether the self-rated knowledge of the positives and negatives associated with air pollution and active transportation shared a relationship with the behaviours taken to limit exposure to air pollution on commutes. Again, individuals that reported lower self-rated knowledge were significantly more likely to report not engaging in any behaviours that would limit exposure to air pollution on commutes, supported with a *p*-value of 0.003.

### 4.6. Thematic Analysis and Health Belief Model

The qualitative results of this survey were interpreted using the Health Belief Model [55] and sought to determine reasoning behind air pollution and health behaviour. 

### 4.7. Perceived Barriers

To evaluate whether air pollution/quality was a perceived barrier to AT, participants were asked what factors influenced their decisions to not use AT. In both groups, participants identified issues relating to distance and time, personal circumstances, and inconvenience as the common reasons for opting out of AT altogether and on certain days for AT users. Primary users of AT were more likely to report weather as a deterrent to active commuting. Unequivocally, neither group referenced air pollution as influencing AT decisions. When asked if active commuters engaged in any behaviours that would limit their exposure to air pollution, few participants identified barriers to engaging in behavioural modifications, such as inefficiency and time incurred while taking alternative routes.

### 4.8. Cues to Action

A series of questions were asked to determine what factors or cues are used to inform health behaviour on commutes. First, participants were asked what steps they take to ensure their safety on active commutes. Responses to this question primarily focused on measures that would prevent immediate or significant bodily harm. When asked more specifically about behaviours adopted to reduce air pollution exposure on commutes, 25% of participants reported adopting some behaviours, however, none mentioned following health messages in the AQHI for behavioural modification. The most reported behaviours for limiting air pollution exposure on commutes included mask-wearing and avoiding busy streets, while some identified avoiding commuting altogether if in the presence of stinky smells and avoiding commuting on hot days.

Evidently, some participants continue to rely on sensory cues or input from the environment to inform their decisions about air quality and their commuting behaviour. These responses highlight suggest an inherent link between heat and poor air quality, with one participant noting “[I] stay indoors when air quality is so bad (in hot weather) to the point where one can’t breathe after doing AT outside for a few minutes”. Another stated, “If a day is hot and has obvious quality issues (industrial odors) I may limit unnecessary errands”. Similar to the previous quote, participants commented on smell as an indicator of poor air quality and used this input as a deciding factor in behavioural modification. This was expressed through responses such as “I avoid factory areas when I smell excessively bad air”, and “I try to avoid the north end of Hamilton when it’s stinky”. Less frequently, visual cues were used to assess air quality with one participant noting “the air *appears* fine in my area”.

### 4.9. Perceived Susceptibility and Perceived Severity (Threat)

Participants were also asked where they perceived air quality to be the worst, and what the primary contributors of air pollution are in Hamilton. Results of the frequency tables revealed that most participants (76%) somewhat agreed or disagreed that air pollution is an issue for them on their commute. The purpose of this question was to determine how perceptions of air quality informs perceptions of susceptibility and severity. Nearly unanimously, participants considered Hamilton’s industrial/steel sector as the site with the poorest air quality and the biggest contributor to pollution. Participants also mentioned areas of high traffic as a contributor to poor air quality and sites of higher pollution. However, no participants identified the complex range of factors (meteorological or topographical) that contribute to the concentration and spread of poor air quality throughout the city beyond the central industrial sector and the major roadways/highways.

### 4.10. Self-Efficacy

When asked about behaviours taken to reduce air pollution exposure on commutes, participants spoke to a lack of perceived control over environmental exposures to air pollution, along with uncertainty of what steps to take to reduce exposure. One participant explicitly noted having constrained autonomy over their personal exposure to air pollution, stating “I don’t have a choice. I live in a polluted area with bad air quality, but I can’t avoid it. Individual behaviour changes don’t amount to much if industry is spewing out toxins all the time”. Similarly, participants noted a lack of awareness that exposure to air pollution can be mitigated on commutes, and what behavioural modifications can be taken. Responses included comments such as, “I didn’t know there were things that could help with this?”, or “To be honest, I probably should, and would be interested to know what others do. It does concern me, but I don’t really know what I can do other than pester the government”. Issues with a perceived self-efficacy in limiting air pollution exposure can reflect earlier findings which noted half of the participants in this survey did not know where to find air quality health information.

## 5. Discussion

This study has demonstrated that the key components of air pollution health literacy, such as the availability, understanding, and utilization of health risk information are not fully developed among active commuters in Hamilton. Overwhelmingly, mitigating air pollution exposure through health-protective behaviours is not a widely adopted practice among active commuters. These findings are consistent with research on AQHI checking in Hamilton [51], which found behavioural modifications in response to health risks to be low. Reasons for low air pollution health literacy are likely due to a correspondingly low perception of health threats introduced by air pollution exposure on active commutes, with the perception of health threats equally contingent on knowledge. EHL [35] posits that an individual’s level of knowledge of the health effects introduced by the environment is a crucial factor in predicting behavioural modification. Participants that reported not checking the AQHI were less likely to engage in any behaviours that would limit their air pollution exposure. As Ramirez et al. [41] noted, the dissemination and quality of air pollution information affect how likely it is that an individual will modify their behaviour, with 50% of the survey sample stating that they do not know where to find the air quality and health information.

Elements of the Health Belief Model, such as cues to action, self-efficacy, and common issues found when evaluating EHL, such as misinformation [35], can further articulate failure among active commutes to engage in air pollution exposure mitigation. Awareness and knowledge of the nature of air pollution in the city of Hamilton could impact cues to action when choosing to adopt health-protective behaviours. Participants primarily reported certain areas, such as Hamilton’s steel industry or main roads, to be the central locale and source of pollutants. While this belief is valid, there appears to be a lack of knowledge regarding the complex factors that contribute to air pollution in the city, and how the meteorological and transboundary characteristics influence the spread of and concentration of air pollutants [44] beyond the areas of public concern. Research by Simone et al. [46] and Howel et al. [47,48] found that perceptions of air quality in Hamilton were bound by geography. As such, the degree of perceived threat based on a geographical context may not elicit cues to actions needed for behavioural modification. Moreover, despite participants identifying busy roadways as sites of air pollution and vehicles as contributors, 78% reported commuting on busy roadways. This finding demonstrates that participants do have some understanding that air quality is an issue when commuting. However, there is a disconnect between perceived susceptibility and severity when responding to this threat, with most active commuters not engaging in any air pollution modifying behaviours at all.

Other points of misinformation that influence cues to action relate to how participants come to understand the nature of air quality. While AQHI consulting was remarkably low, the qualitative data in this survey suggested that some participants rely on environmental or sensory cues to determine whether air quality is poor and if behavioural modification is necessary. Specifically, individuals rely on heat or hot temperatures as a cue to action for modifying their behaviour in the presence of air pollution. This belief is misleading, as concentration of pollutants in the air is not directly determined by heat and is thus not a reliable cue for determining when engaging in health-protective behaviour is necessary. Similarly, participants noted modifying their behaviours if they detected odor, as this characteristic was thought to share a relationship with air pollution. While some pollutants do produce an odor, many of the most concerning pollutants and gases to public health are odorless. These findings are consistent with research conducted by Elliot et al. [49] which reported similar sensory cues employed by participants, such as smells, heat, and visual inputs to evaluate when the air quality is poor.

Lastly, participants expressed having little self-efficacy with respect to perceived ability to limit air pollution exposure. Self-efficacy, as detailed in the Health Belief Model, is critical for determining whether or not an individual will participate in health-protective behaviours [54]. The level of perceived control an individual has in mitigating environmental health risks circles back to deficits in critical knowledge needed to make informed and actionable health decisions. Lindsey et al. [38] stress the importance of EHL communication that is action-oriented and comprehendible among the lay public to ensure its efficacy. Despite the underutilization of current air quality health tools, such as the AQHI, there was general agreeance among the survey sample that information on the health impacts of air pollution on active commutes should be made more available to the public.

## 6. Conclusions

There is a well-established relationship between air pollution exposure and adverse health outcomes. Health risks incurred through exposure to air pollution on active commutes places active commuters at an increased risk of experiencing adverse health outcomes through various processes, such as high inhalation rates, proximity, duration, and time. However, how active commuters come to understand this vulnerability, and whether they engage in any behaviours to mitigate air pollution exposure has been under-explored. This research sought to fill this gap by using the Environmental Health Literacy (EHL) framework to evaluate how health-risk information is accessed, understood, and mobilized among active commuters. It was revealed that air pollution is an often-neglected consideration during active travel, with most active commuters not engaging in any behaviours that would limit exposure (75%), such as AQHI checking and following health messages. These findings are consistent with the literature on the EHL framework, which argues that accuracy and comprehension of information are critical for estimating behavioural modification and personal environmental risk assessment.

While this research revealed important insights concerning active commuters’ knowledge of health risks associated with air pollution on commutes, there are some limitations that should be noted. This research was based on participant recruitment through voluntary sampling, thereby introducing several issues with representativeness, despite efforts to recruit a diverse and robust sample. Moreover, while the sample size is smaller than what is typically standard for quantitative research, precautions were taken to ensure the target audience was selected for participation. Unequivocally, as this study focused on air pollution knowledge among active commuters in a specific locale (Hamilton, ON), these results cannot be directly generalized to other areas/populations. Evaluating air pollution health literacy with a larger sample size and in different cities can further contribute to the literature on this topic. Online surveys are also more likely to experience participant attrition, and as is true for any research dealing with human participants, there is the chance of these results being informed by recall or social desirability bias. Importantly, this research was not able to speak to any correlations between air pollution health literacy and demographics/socioeconomic status. Lastly, due to the availability of air quality and health information, this research assumed active commuters understood themselves as an at-risk group; however, the results argue that this may not be the case. As such, this research was not able to make conclusions about whether active commuters identify as an at-risk group, or what factors/characteristics they would associate with air pollution health risk (i.e., personal health status, co-morbidities, age etc.).

Conversely, this research has extended the established relationship between knowledge and behaviour, to active commuters, providing novel insights into how this risk-group understands and engages with air pollution health information. It has also been attempted to provide a base survey tool for evaluating air pollution health literacy that can be replicated and/or adapted for future research.

In addition to these contributions, other air quality and health measures should be addressed as cities transition towards more sustainable forms of transportation. Notably, active commuters do not only have deficits in air pollution knowledge, but many also do not know where to access air pollution and health information at all. While air pollution and health information are available to the public, this research has revealed that engagement with it is low. Here, the key components of the Health Belief Model can be used to create cues to action among the public and elicit engagement with air quality-checking tools. This could include air quality advertisements that are visible on AT routes. Importantly, air pollution communication that is both effective and comprehensible has been proven to be effective in creating behavioural changes.

Moreover, a lack of critical information on the nature of the air quality in the city of Hamilton stresses the importance of approaches to improving air pollution health literacy that are adaptive to specific locational contexts. Further outreach is required from local governments and agencies/groups with the goal of improving the knowledge of the main sources and spread of air pollution throughout the city, and how exposure to air pollution can have both immediate and long-term health implications. The use of air quality campaigns, information sessions, and community engagement could serve as appropriate avenues for furthering air pollution health literacy among the active commuting population.

Undoubtedly, as vehicle traffic is a primary contributor to poor air quality, continued efforts to transition from vehicular to sustainable modes of transportation is recommended. Various stakeholders can work to encourage this use of AT, such as government incentives for using AT, flexible work times to avoid rush hour pollution, and AT priority corridors. Similarly, urban and transportation planning policy and research would benefit from participatory research when planning AT routes that are efficient and consider mitigated air pollution exposure. These efforts are integral to improving the air pollution health literacy among active commuters in Hamilton and working towards maximizing the health benefits for active commuters.

## Figures and Tables

**Figure 1 ijerph-20-06282-f001:**
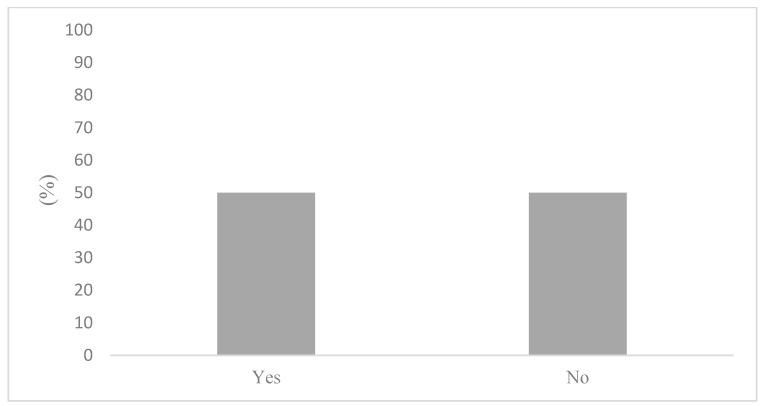
Knowing where to find air quality and health information.

**Figure 2 ijerph-20-06282-f002:**
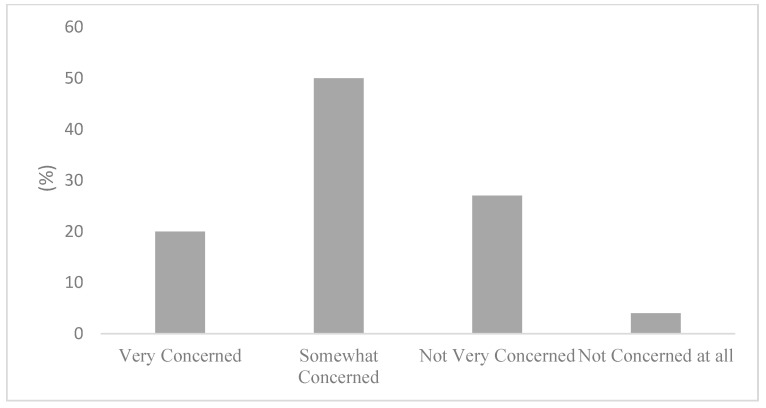
Concern of health risks associated with air pollution.

**Figure 3 ijerph-20-06282-f003:**
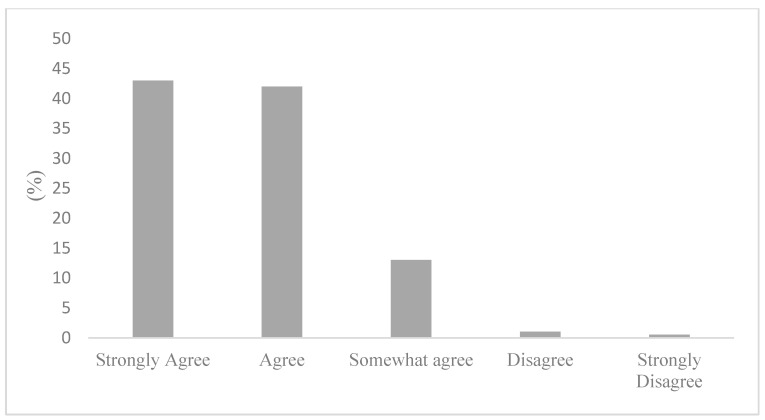
Information on air pollution exposure during active commutes should be more available to the public.

**Figure 4 ijerph-20-06282-f004:**
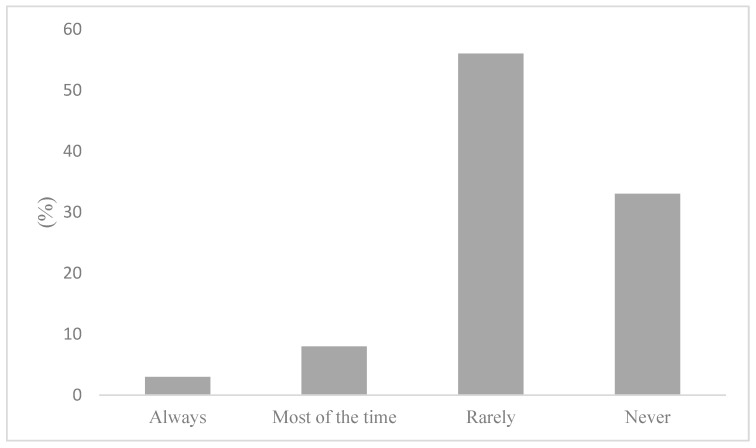
Air quality informing active transportation decisions.

**Figure 5 ijerph-20-06282-f005:**
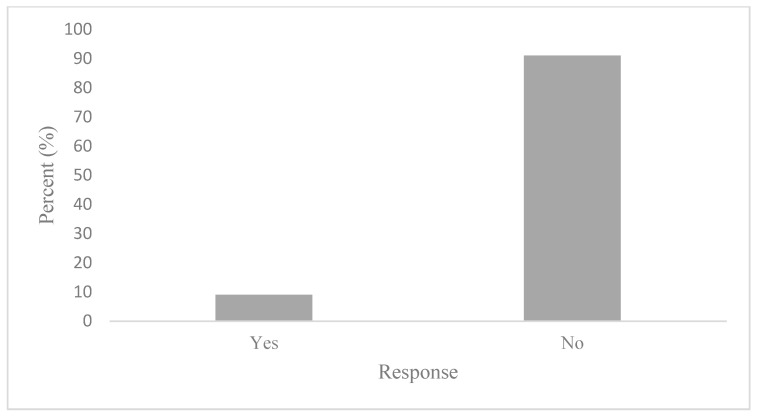
Daily AQHI checking.

**Figure 6 ijerph-20-06282-f006:**
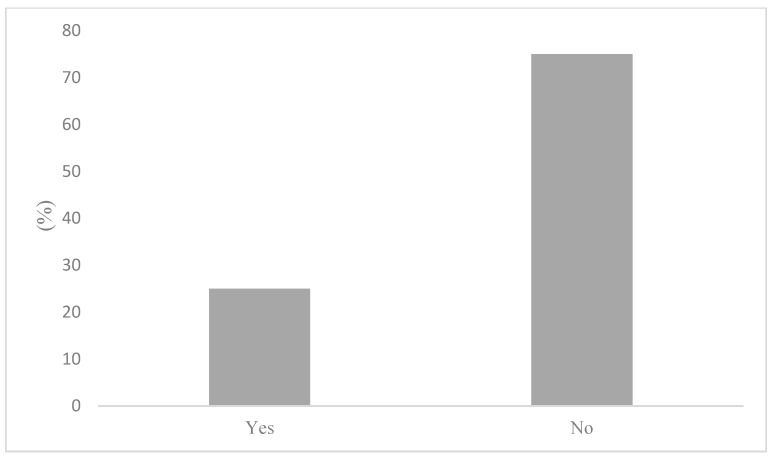
Engaging in behaviour to mitigate pollution exposure.

**Table 1 ijerph-20-06282-t001:** Air quality knowledge and health behavior.

	Do you engage in any behaviours that would limit your exposure to air pollution during your commute?			
		No	Yes	Total
Do you consult the air quality health index (AQHI) prior to your active commutes/travels daily?	No	124 (79%)	33 (21%)	157 (100%)
	Yes	5 (31.2%)	11 (68.8%)	16 (100%)
Total		129 (74.6%)	44 (25.4%)	173 (100%)
x^2^ = 15.017 df = 1 ǫ = 0.318 Fisher’s *p* = 0.000				

**Table 2 ijerph-20-06282-t002:** Air quality knowledge and health behavior.

	On a scale of 1–5, how would you rate your knowledge of the positives and negatives associated with air pollution and active transportation? Where 5 is very knowledgeable and 1 is not knowledgeable at all				
Do you know where to seek information regarding air quality and health?		1	3	5	Total
	No	42 (47.7%)	33 (37.5%)	13 (14.8%)	88 (100%)
	Yes	30 (34.5%)	30 (34.5%)	27 (31%)	87 (100%)
Total		74 (41.3%)	63 (35.2%)	42 (23.5%)	179 (100%)
x^2^ = 10.746 df = 4 Cramer’s V = 0.249 Fisher’s *p* = 0.027					

**Table 3 ijerph-20-06282-t003:** Air quality knowledge and health behavior.

	Do you engage in any behaviours that would limit your exposure to air pollution during your commute?			
		No	Yes	Total
On a scale of 1–5, how would you rate your knowledge of the positives and negatives associated with air pollution and active transportation? Where 5 is very knowledgeable and 1 is not knowledgeable at all	1	65 (87.8%)	9 (12.2%)	74 (100%)
	3	43 (68.3%)	20 (31.7%)	63 (100%)
	5	26 (61.9%)	16 (38.1%)	42 (100%)
Total		134 (74.9%)	45 (25.1%)	179 (100%)
x^2^ = 11.829 df = 2 Cramer’s V = 0.257 Fisher’s p = 0.003				

## Data Availability

The raw data is not available upon request, or publicly available as it would breach the ethics and participant confidentiality agreement. Aggregates of the data are presented within the article.

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
