# Peer review of "Air Pollution Health Literacy among Active Commuters in Hamilton, Ontario"

_ijerph, 2023, doi:10.3390/ijerph20136282_

Round 1

Reviewer 1 Report

Review for “Air Pollution Health Literacy among Active Commuters in Hamilton, Ontario”

The researchers set out to survey active transportation commuters to evaluate their air pollution health literacy in terms of health risks and behaviors to mitigate those risks.  They conclude that the majority of AT commuters’ knowledge is low, and air pollution appears to be a neglected consideration.  Consequently, AT commuters do not adopt behaviors that would limit air pollution exposure.

Overall, I like this paper and believe if makes an important contribution.  It is well written and organized.  I do have some questions and recommendations that I believe can further strengthen the paper.  Specifically, the paper, in its current form, lacks context in terms of Hamilton’s air pollution problems, causes, etc. to better understand the study’s results and where to go next.  My comments below strive to help the researchers provide further context for the study. 

Abstract:  The term “active transportation” needs to be defined in the abstract so that readers unfamiliar with the term can understand the focus of the study.

2.6:  Factors driving Hamilton’s poor air quality need to be better developed and discussed earlier in the paper.  Given that this is a case study about Hamilton (as noted in the paper’s title), the context of why the researchers are studying AT commuter air pollution literacy for this city warrants more than one sentence (lines 193-195).  Issues contributing to the pollution can help readers better understand how Hamilton residents view their local air pollution, causes, and if causes can be mitigated (or perceived to be mitigated).  In Utah, for example, poor air quality is due to winter inversions that trap auto and building emissions in bowl-like mountain valleys.  Frustratingly, because of Utah’s typography, many residents believe there isn’t much they can do to address local winter pollution.  Details, such as this, about Hamilton residents can help the reader better understand circumstances and how residents view and respond to their air pollution.  Later in the paper, the researchers assert that “heat” and other factors impact AT commuter perceptions of pollution.  Having a better understanding what is contributing to Hamilton’s air pollution can help the reader better discern the context of these results.

4.4.2:  Why do AT commuters believe heat is a factor contributing to air pollution?  Explain.  Does heat contribute to ozone?  Do coal-fired power plants spew more/visible pollutants to power air conditioning?  Again, details are needed when discussing these results.

4.4.3:  Related to the above, are their studies that identify the actual breakdown of specific contributors to air pollution?   How much does the industrial steel mill contribute?  What about cars?  What about use of heating in buildings?  What contributes to the smell of the pollution?  Articulation of the contributors to local air pollution can help the reader better understand how local residents view their problem.

7.  The most important issue that warrants further development in this paper is proposing specific strategies for educating the public about air pollution.  The authors find that AT commuters lack important knowledge about air pollution and appropriate behaviors to address it.  Which education gaps are the most important?  How do the researchers propose that Hamilton residents fill those gaps?  Given how difficult it is to educate adults on environmental issues given that there are few opportunities to school them in mass, what do the researchers recommend?  Are public health agencies the only ones responsible?  What about schools and clean air advocacy groups?   Can kids learning about pollution in schools in turn teach their parents?  What is government’s role? In short, what insights from the researchers’ results point to specific directions for education? 

Education appears to be the next important step in this research, and the researchers should help articulate the next directions for others to investigate based on their results.

Author Response

Reviewer 1: Thank you for taking the time to review our manuscript and for your thoughtful comments. We have worked to edit the manuscript as best as possible based on your feedback. Below you will find the responses to your comments and the action we took to improve the manuscript.

Abstract:  The term “active transportation” needs to be defined in the abstract so that readers unfamiliar with the term can understand the focus of the study.

Response 1: The abstract was edited to include a brief definition of active transportation (pg.1)

2.6:  Factors driving Hamilton’s poor air quality need to be better developed and discussed earlier in the paper.  Given that this is a case study about Hamilton (as noted in the paper’s title), the context of why the researchers are studying AT commuter air pollution literacy for this city warrants more than one sentence (lines 193-195).  Issues contributing to the pollution can help readers better understand how Hamilton residents view their local air pollution, causes, and if causes can be mitigated (or perceived to be mitigated).  In Utah, for example, poor air quality is due to winter inversions that trap auto and building emissions in bowl-like mountain valleys.  Frustratingly, because of Utah’s typography, many residents believe there isn’t much they can do to address local winter pollution.  Details, such as this, about Hamilton residents can help the reader better understand circumstances and how residents view and respond to their air pollution.  Later in the paper, the researchers assert that “heat” and other factors impact AT commuter perceptions of pollution.  Having a better understanding what is contributing to Hamilton’s air pollution can help the reader better discern the context of these results.

Response 2: A case study section was added to the methodology (formerly 2.6) where more information was provided regarding the nature and contributors of air pollution within the city of Hamilton. This should provide the reader with better context for the results in the latter section as well (i.e. air pollution) (pg.5)  

4.4.2:  Why do AT commuters believe heat is a factor contributing to air pollution?  Explain.  Does heat contribute to ozone?  Do coal-fired power plants spew more/visible pollutants to power air conditioning?  Again, details are needed when discussing these results.

Response 3: Participants voluntarily disclosed their perceptions of air pollution to be associated with heat (i.e. assumed that when it is hot outside the air quality would be poor). They were not asked a direct question about heat and air quality and as such were not asked to expand on why they held this belief. 

4.4.3:  Related to the above, are their studies that identify the actual breakdown of specific contributors to air pollution?   How much does the industrial steel mill contribute?  What about cars?  What about use of heating in buildings?  What contributes to the smell of the pollution?  Articulation of the contributors to local air pollution can help the reader better understand how local residents view their problem.

Response 4: The new case study sections provide more information/ context on the nature of air pollution in Hamilton to help the reader understand some of the responses in the results (pg.5)

  1. The most important issue that warrants further development in this paper is proposing specific strategies for educating the public about air pollution. The authors find that AT commuters lack important knowledge about air pollution and appropriate behaviors to address it.  Which education gaps are the most important?  How do the researchers propose that Hamilton residents fill those gaps?  Given how difficult it is to educate adults on environmental issues given that there are few opportunities to school them in mass, what do the researchers recommend?  Are public health agencies the only ones responsible?  What about schools and clean air advocacy groups?   Can kids learning about pollution in schools in turn teach their parents?  What is government’s role? In short, what insights from the researchers’ results point to specific directions for education? Education appears to be the next important step in this research, and the researchers should help articulate the next directions for others to investigate based on their results.

Response 5: The conclusion section was edited to further development a conversation of future research directions and how air pollution health literacy can be addressed within the city of Hamilton among the active commuter community. (pg. 16-17)

Reviewer 2 Report

The research explores the behaviour of actively mobile commuters in order to test their understanding of the health risks associated with air pollution and the possible subsequent adoption of preventive measures.
The article is therefore fully in line with the aims of the journal. 
Although it deals with an interesting and topical subject and is therefore worthy of dissemination, the article needs careful revision in terms of structure and presentation of the results.

More in detail, here are some comments.

In the Kwords, I would add Active Travel (a useful synonym for expanding literature and justifying public choice); 
TO T. Litman certainly deserves some more defining merit.

I would add some references from transport literature; specifically I would add some sources to the statements in the introduction 
(see, for example, for definition - Cook, S., Stevenson, L., Aldred, R., Kendall, M., & Cohen, T. (2022). More than walking and cycling: What is 'active travel'?. Transport Policy, 126, 151-161.
While on lines 32/33 , specify urban scale and recent times  Maltese, I., Gatta, V., & Marcucci, E. (2021). Active travel in sustainable urban mobility plans. An Italian overview. Research in Transportation Business & Management, 40, 100621.

I would also mention the issue of safety, since an increase in active travel that does not correspond to a certain decrease in motorised traffic also exposes active travellers to this other risk.
Therefore, it would be useful to briefly mention the motivations of the study, i.e. on the one hand why the authors chose to focus on this aspect of the problem of unsustainable mobility and on the commuter category (as obvious as these considerations may seem, they are important from a scientific approach point of view); on the other hand, what is the added value of this publication with respect to the status quo of knowledge on the topic, but also with respect to the possible policy implications that may result from it.
Also missing is some necessary mention of the methods of analysis that will be specified later, namely the questionnaire for data collection in the chosen area and the models used for data analysis.
At the end of the introductory section, I would add a description of the structure of the entire article.

At the beginning of section two, I would add a sentence introducing what subsections follow and what the purpose of the section itself is. Indeed, it is not clear whether the intention is to give the current state of the literature on the subject or to describe the different aspects related to TA. If it is the former, then the section should be restructured into a description of what certain authors have said about certain topics (as has been done in subsections 2.4 and 2.5). If it were instead the second option, then the first three subsections would have to be better defined as they appear partially overlapping.
I would move section 2.6 to methodology - section three - titling it something like 'case study description'.

In section three I would add the introductory sentence at the beginning and why the specific geographical locus was chosen. In the definition of primary I would specify whether the 50% refers to the number of modes used, time or distance. Since these are commuting trips at least from the title explain how the respondents were chosen (beyond the mention of bicycle shops it must be understood why they were chosen to assist the research), whether commuting to school is also considered commuting, whether TA is considered as utilitarian and possibly included in trip chaining.

Check section three, erroneously repeated -data analysis- because it is clearly not long enough to be a section; it could easily be incorporated into the previous one, I would say, where the description of the HBM model is also missing, which should be better described (just take and comment on figure 1, anticipating it...) and attributed to a certain scientific background as well as to specific authors, one supposes. In addition, the software used is very irrelevant to the type of analysis conducted; descriptive statistics? That's all very well, but it has to be said.

Section 4 of the results should also be briefly introduced at the beginning by specifying that the subsequent subsections retrace the structure between separate sections of the questionnaire submitted to voluntary respondents and are distinguished according to quantitative or qualitative analysis. 
Check the numbering of the subsections 
4.2 looks like 4.1.2? 
Tables one and two may well be moved to the appendix as they do not add much to what is said in the commentary (which I would expand on at most); alternatively they should certainly be placed in the reference section then as 4.1.1 and 4.1.2 respectively
It is not clear why there is talk of quantitative analysis only at 4.3 in fact that is also the case at the previous subsection precisely to review then headings 4. 3 and the appropriateness of this subdivision; in any case, table 3 could be better represented with graphs and only for some of the questions; again the description of what is asked in the questionnaire and the pointing out of all the percentages would be better in the appendix - in any case the questions should be numbered and in the text the comment that should be enriched should refer to the number of the question in the table or graph: here I would focus on the answers and the most significant results with an effort of greater synthesis.
Instead of distinguishing between frequencies and cross tabulation I would better specify why it is decided to analyse the correlation between specific answers.
ATTENTION to the title: The analysis is always quantitative - at the limit are the questions that require qualitative answers: take this into account when revising the section titles.

In general for the future, remember not only to number the questions, but also to put those on knowledge of a topic before those on behaviour, while those on the socio-demographic nature of the respondents always go at the end.

I would not devote a whole section to the limitations of this work; instead I would simply mention them in the conclusions (so as to give even less prominence to the limitations themselves precisely...); at the same time I would also suggest further directions of research that can contribute to overcoming these limitations: in this way the solution to the problem is already shown

In the conclusions I would always refer to the fact that I am talking about a specific city and its inhabitants, handling any generalisations with caution.

Author Response

Reviewer 2: Thank you for taking the time to review our manuscript and for your thoughtful comments. We have worked to edit the manuscript as best as possible based on your feedback. Below you will find the responses to your comments and the action we took to improve the manuscript.

In the Kwords, I would add Active Travel (a useful synonym for expanding literature and justifying public choice);

Response 1: Active Travel was added to keywords (pg.1)

TO T. Litman certainly deserves some more defining merit.

I would add some references from transport literature; specifically I would add some sources to the statements in the introduction (see, for example, for definition - Cook, S., Stevenson, L., Aldred, R., Kendall, M., & Cohen, T. (2022). More than walking and cycling: What is 'active travel'?. Transport Policy, 126, 151-161.

Response 2: Cook et al., reference now reflects the utilized definition of active travel in this paper. Litman was incorporated into the introduction (pg. 1)

While on lines 32/33 , specify urban scale and recent times  Maltese, I., Gatta, V., & Marcucci, E. (2021). Active travel in sustainable urban mobility plans. An Italian overview. Research in Transportation Business & Management, 40, 100621.

Response 3: this reference was used to support comments surrounding safety and pollution in urban areas. (pg.1 &2)

 I would also mention the issue of safety, since an increase in active travel that does not correspond to a certain decrease in motorised traffic also exposes active travellers to this other risk. Therefore, it would be useful to briefly mention the motivations of the study, i.e. on the one hand why the authors chose to focus on this aspect of the problem of unsustainable mobility and on the commuter category (as obvious as these considerations may seem, they are important from a scientific approach point of view); on the other hand, what is the added value of this publication with respect to the status quo of knowledge on the topic, but also with respect to the possible policy implications that may result from it.

Response 4: Further discussion of safety on commutes and the motivations for this study (air pollution) were added or made clearer within the introductory paragraph. A statement was also added to reflect the significance of this research to the current literature and policy (pg.2)

Also missing is some necessary mention of the methods of analysis that will be specified later, namely the questionnaire for data collection in the chosen area and the models used for data analysis.

At the end of the introductory section, I would add a description of the structure of the entire article.

Response 5: An additional paragraph was added to the introduction to outline the structure and following sections of the paper (pg.2)

At the beginning of section two, I would add a sentence introducing what subsections follow and what the purpose of the section itself is. Indeed, it is not clear whether the intention is to give the current state of the literature on the subject or to describe the different aspects related to TA. If it is the former, then the section should be restructured into a description of what certain authors have said about certain topics (as has been done in subsections 2.4 and 2.5). If it were instead the second option, then the first three subsections would have to be better defined as they appear partially overlapping.

Response 6: A sentence was added at the beginning of section 2, to reflect our focus on the current knowledge surrounding air pollution and health on active commutes. Sub-headings were also altered to more clearly define the purpose of each sub-section (pg. 2-5)

I would move section 2.6 to methodology - section three - titling it something like 'case study description'.

In section three I would add the introductory sentence at the beginning and why the specific geographical locus was chosen.

Response 7: Section was moved into the methodology and reframed as a case study. Further contextual information was added to the case study to explain why Hamilton was the focus. (pg.5)

In the definition of primary I would specify whether the 50% refers to the number of modes used, time or distance. Since these are commuting trips at least from the title explain how the respondents were chosen (beyond the mention of bicycle shops it must be understood why they were chosen to assist the research), whether commuting to school is also considered commuting, whether TA is considered as utilitarian and possibly included in trip chaining.

Response 8: Further explanation was provided re: definition used in this study for exclusionary/inclusionary criteria of AT and how recruitment for the study was conducted (pg.6-7)

Check section three, erroneously repeated -data analysis- because it is clearly not long enough to be a section; it could easily be incorporated into the previous one, I would say, where the description of the HBM model is also missing, which should be better described (just take and comment on figure 1, anticipating it...) and attributed to a certain scientific background as well as to specific authors, one supposes.

Response 9: Data analysis section was incorporated into results and explanation and purpose of the HBM was expanded and referenced using the literature.  (pg.7)

In addition, the software used is very irrelevant to the type of analysis conducted; descriptive statistics? That's all very well, but it has to be said.

Response 10: descriptive statistics added (pg.7)

Section 4 of the results should also be briefly introduced at the beginning by specifying that the subsequent subsections retrace the structure between separate sections of the questionnaire submitted to voluntary respondents and are distinguished according to quantitative or qualitative analysis.

Response 11: a statement at the beginning of the results section was added to reflect the structuring of the section. (pg.7)

Check the numbering of the subsections

4.2 looks like 4.1.2?

Response 12: corrected numbering of sections (pg.7-8)

Tables one and two may well be moved to the appendix as they do not add much to what is said in the commentary (which I would expand on at most); alternatively they should certainly be placed in the reference section then as 4.1.1 and 4.1.2 respectively

Response 13: Tables were moved to appendix (pg. 17)

It is not clear why there is talk of quantitative analysis only at 4.3 in fact that is also the case at the previous subsection precisely to review then headings 4. 3 and the appropriateness of this subdivision; in any case, table 3 could be better represented with graphs and only for some of the questions; again the description of what is asked in the questionnaire and the pointing out of all the percentages would be better in the appendix - in any case the questions should be numbered and in the text the comment that should be enriched should refer to the number of the question in the table or graph: here I would focus on the answers and the most significant results with an effort of greater synthesis.

Response 14: Table three was moved to the appendix. Graphs were created to address the most salient responses, discussion of the results was added. (pg. 8-10)

Instead of distinguishing between frequencies and cross tabulation I would better specify why it is decided to analyse the correlation between specific answers.

Response 15: explanation of why correlation was chosen was added (pg.11)

ATTENTION to the title: The analysis is always quantitative - at the limit are the questions that require qualitative answers: take this into account when revising the section titles.

Response 16: titled removed

In general for the future, remember not only to number the questions, but also to put those on knowledge of a topic before those on behaviour, while those on the socio-demographic nature of the respondents always go at the end.

Response 17: questions were numbered and presentation of the descriptive results was re-ordered so that knowledge-based results are discussed first (pg. 9-10)

I would not devote a whole section to the limitations of this work; instead I would simply mention them in the conclusions (so as to give even less prominence to the limitations themselves precisely...); at the same time I would also suggest further directions of research that can contribute to overcoming these limitations: in this way the solution to the problem is already shown

In the conclusions I would always refer to the fact that I am talking about a specific city and its inhabitants, handling any generalisations with caution.

Response 18: The limitations section has been consolidated into the conclusions section and a comment was added referring to generalizations of the research findings considering the locale of interest. Also, the conclusion section was edited to provide further development of future research directions and how air pollution health literacy can be addressed within the city of Hamilton among the active commuter community and how to overcome the limitations addressed. (pg. 15-16)

Reviewer 3 Report

The subject treated by the paper is interesting, but there are certain weaknesses in the methodological part. The mixed-method survey concept is not clear. Please make reference in the literature at this method. Also, for the quantitative research, the sample size and the sampling method are not likely to lead to relevant results . The bivariate analysis is precarious, consisting in only one crosstabulation evaluation.  I  recommend the usage of more elaborate  methods of analysis for the quantitative data . For the qualitative data, the thematic analysis should be explained. The introduction has to present the paper structure. 

The English language requires minor editing.

Author Response

Reviewer 3: Thank you for taking the time to review our manuscript and for your thoughtful comments. We have worked to edit the manuscript as best as possible based on your feedback. Below you will find the responses to your comments and the action we took to improve the manuscript.

The mixed-method survey concept is not clear. Please make reference in the literature at this method.

Response 1: the methodology was more clearly defined in section 3 & 4 (pg. 5-7)

Also, for the quantitative research, the sample size and the sampling method are not likely to lead to relevant results .

Response 2: a comment was added to the conclusion commenting on the limitations of the sample size and sampling method. (pg.15-16)

The bivariate analysis is precarious, consisting in only one crosstabulation evaluation.  I  recommend the usage of more elaborate  methods of analysis for the quantitative data .

Response 3: Additional crosstabulations were added to the results to further develop the conversation on knowledge and behaviour (pg. 11-12)

For the qualitative data, the thematic analysis should be explained.

Response 4: thematic analysis was defined utilizing the literature  (pg.7)

The introduction has to present the paper structure.

Response 5: the introduction was edited to reflect the paper structure and its contents (pg. 2)

Round 2

Reviewer 2 Report

I thank the authors for following my comments and suggestions. 
I think the article is ready to be published as long as a couple of very minor things are fixed.

Firstly, as already previously suggested, there should not be two adjacent titles so whenever a section has a series of subsections it is a good idea for there to be a couple of introductory lines saying section X is divided into section X1 which talks about ------- and section x2 which talks about -------. 
Another notation: figure captions must go to the substance not refer to numbered questions from a questionnaire we don't even know.
Take graph 1 - which is also to be called figure 1 -: it should be renamed something like 'finding information on air quality and health'. 
I would eliminate the titles inserted within the figure itself, leaving more space for the Cartesian axes, and I would avoid heading the y-axis when simply adding the %; the reader can easily understand what you are referring to. The same applies to the x-axis where writing answers is quite unnecessary, since it is clear what the figures represent, i.e. the answers given in the survey...

Similarly, if section 4.2 is called cross tabulation, it is pointless to rewrite it each time in the title of the tables, the focus of which would instead be on the elements being related.

Author Response

Reviewer 2: Thank you again for your detailed review comments. Your feedback has helped improve the quality of our manuscript. We have worked to address the comments as best as possible.   

Firstly, as already previously suggested, there should not be two adjacent titles so whenever a section has a series of subsections it is a good idea for there to be a couple of introductory lines saying section X is divided into section X1 which talks about ------- and section x2 which talks about -------.

Response 1: Multiple titles were removed and a more explicit discussion of the subsections was added to the introductory paragraph in the Results section.

Another notation: figure captions must go to the substance not refer to numbered questions from a questionnaire we don't even know. Take graph 1 - which is also to be called figure 1 -: it should be renamed something like 'finding information on air quality and health'.

Response 2: Graph labeling changed to figure. Titles were changed to reflect the contents of the graph.

I would eliminate the titles inserted within the figure itself, leaving more space for the Cartesian axes, and I would avoid heading the y-axis when simply adding the %; the reader can easily understand what you are referring to. The same applies to the x-axis where writing answers is quite unnecessary, since it is clear what the figures represent, i.e. the answers given in the survey...

Response 3: Titles were removed from the graphs themselves, and more descriptive titles were given to the graphs as discussed in the previous comment

Similarly, if section 4.2 is called cross tabulation, it is pointless to rewrite it each time in the title of the tables, the focus of which would instead be on the elements being related.

Response 4: Titles removed from within crosstabulations and table title changed

Reviewer 3 Report

The authors took into account the recommendations and they have improved the paper.

Author Response

Reviewer 3: Thank you again for your review comments. Your feedback has helped improve the quality of our manuscript.